# Adapted or Adaptable: How to Manage Entropy Production?

**DOI:** 10.3390/e22010029

**Published:** 2019-12-24

**Authors:** Christophe Goupil, Eric Herbert

**Affiliations:** Université de Paris, Laboratoire Interdisciplinaire des Energies de Demain (LIED), UMR 8236 CNRS, F-75013 Paris, France; eric.herbert@univ-paris-diderot.fr

**Keywords:** out of equilibrium thermodynamics, finite time thermodynamics, living systems

## Abstract

Adaptable or adapted? Whether it is a question of physical, biological, or even economic systems, this problem arises when all these systems are the location of matter and energy conversion. To this interdisciplinary question, we propose a theoretical framework based on the two principles of thermodynamics. Considering a finite time linear thermodynamic approach, we show that non-equilibrium systems operating in a quasi-static regime are quite deterministic as long as boundary conditions are correctly defined. The Novikov–Curzon–Ahlborn derivation applied to non-endoreversible systems then makes it possible to precisely determine the conditions for obtaining characteristic operating points. As a result, power maximization principle (MPP), entropy minimization principle (mEP), efficiency maximization, or waste minimization states are only specific modalities of system operation. We show that boundary conditions play a major role in defining operating points because they define the intensity of the feedback that ultimately characterizes the operation. Armed with these thermodynamic foundations, we show that the intrinsically most efficient systems are also the most constrained in terms of controlling the entropy and dissipation production. In particular, we show that the best figure of merit necessarily leads to a vanishing production of power. On the other hand, a class of systems emerges, which, although they do not offer extreme efficiency or power, have a wide range of use and therefore marked robustness. It therefore appears that the number of degrees of freedom of the system leads to an optimization of the allocation of entropy production.

## 1. Introduction

The issue of energy conversion is the subject of historical debate. Without going back to its roots, let us mention the work initiated by Glansdorf and Prigogine, which placed at the center the question of entropy production in out-of-equilibrium systems, an issue that is still largely relevant [1,2]. This debate is itself part of an even broader debate that questions the operating points of the systems, considering mainly the maximization of entropy production (MEP), its minimization (mEP), or power maximization (MPP) [3,4]. One of the reasons why these questions do not find a general consensus today is that they are most often considered on very different systems, in particular in the definition of the boundary conditions of the device with its environment, considered immutable. The case of idealized mechanical systems is, from this point of view, much simpler, since, broadly speaking, the absence of any friction process means that the system interacts with its environment via a very limited number of degrees of freedom, which makes variational approaches relevant. On the contrary, it has long been accepted that there is no variational principle that governs the out-of-equilibrium steady state of a thermodynamic system [5]. This can be understood as an impossibility to establish a variational principle when the number of degrees of freedom diverges, which is obviously the case when the system is connected to a thermostat, and when dissipative processes occur. However, it is equally obvious that many out-of-equilibrium systems are perfectly deterministic in their evolution, and have a perfectly defined stationary state, as is the case, for example, for Kirchoff’s networks in electronics. As a result, these systems, although not governed by a Lagrangian form and an associated variational principle, have a completely established stationary operating point, without any possible affirmation of an underlying minimization or maximization of the production of the entropy or the power.

These questions of power and finite time performance have been the subject of much work [6] particularly in thermoelectricity [7,8,9,10,11]. Without entering into these debates again, we propose an approach that provides a fairly generic framework for describing a complete thermodynamic system with perfectly established boundary conditions. In this article, we will limit ourselves to the case of locally linear machines, subscribing to Onsager’s formalism. This formalism, based on the concept of local equilibrium, makes it possible to consider the thermodynamic potentials of the system, which are the intensive parameters. As a result, it becomes possible to derive a thermodynamics close to equilibrium, with, in particular, a rigorous choice of potentials that allow for obtaining the symmetry of the out-of-diagonal coefficients of the Onsager matrix. The stationary nature also requires that kinetic coefficients and boundary conditions of the system be constant or slowly variable compared to the characteristic relaxation time of entropy production and dissipation diffusion, thus guaranteeing both stationary processes and local equilibrium.

In this article, we consider the transport of energy and matter within a system, where the thermodynamic conversion is produced by coupling the energy and matter currents. By applying the first law of thermodynamics, both of these currents are conservative. By applying the second law, the energy, and sometimes the matter, used during the conversion process is subject to dispersion in the degrees of freedom accessible to the system. As a result, thermodynamics is based on both quantity and quality principles. Since the loss of quality is directly related to dispersion in the degrees of freedom, the search for processes to reduce their number has always been a guideline. It should be noted that, in the case of non-spontaneous processes, it is possible to consider a reduction in the degrees of freedom, but this operation requires the implementation of external processes. These processes offer other opportunities for energy dispersion, in greater proportions than those gained within the system. As a result, any physical process taking place over a finite period of time is the location of a compromise between the total energy used to carry out a process, and the energy actually converted for the needs to be covered. The process efficiency is therefore written as the ratio between the actually converted energy and the total energy supplied. We propose to consider energy conversion processes in a very generic form, in order to establish their main characteristics and constraints. In particular, we address the question of power and entropy production, insisting on the compromises they impose.

The question of adapting a device to the uses assigned to it then arises. In the case of single working point, the system may be designed to be as much adapted as is it possible. However, this single operating working point is a rare configuration, and realistic systems are asked to work in a given range of working points. Then, the concept of adaptability, or flexibility, arises, which enters into competition with the previous adapted concept. This problem of adaptation or adaptability concerns all thermodynamic systems, including, of course, living systems. Indeed, as soon as we define an envelope, we delimit the boundaries of a space occupied by a given device and the interactions of this device with the outside world. Considering the energy and matter budget at the borders of the device, we then characterize the relationship between the device and its environment. Since the processes take place over a finite period of time, it is important to consider an out-of-equilibrium description. In this paper, we consider an out-of-equilibrium thermodynamic description, driven by locally linear equations. We show that the intrinsic characteristics of the device, on the one hand, and the boundary conditions, on the other hand, totally determine the behavior of the system. It appears that the allocation of dissipation largely determines the possible ranges of use of an out-of-equilibrium thermodynamic system.

In terms of boundary conditions, we show that the real coupling conditions of a system with its environment are always located between the Dirichlet and Neumann boundaries, also called “stock” and “flow” boundary conditions. It should be noted that both pure stock and flow are extreme boundary conditions which can never being strictly reached. Between adaptable and adapted, the performances of thermodynamic systems are therefore the result of a compromise between intrinsic performance of a device and the coupling to the environment. This question of coupling to the environment is the subject of the first section of this article. In the following section, we describe the envisaged system in its most general form. The third section concerns the descriptions of the device at the heart of the system, while the fourth section describes its insertion into the complete system. The fifth section considers the different configurations that such a global system may encounter, and the consequences on the production of power, dissipation, and more generally, entropy. The article ends with concluding remarks.

## 2. System Description

### 2.1. Boundary Conditions

As indicated above, the system is composed of two sub-parts: a central zone, which we will call the device, and which is the place of thermodynamic conversion, on the one hand, and the boundary conditions, consisting of the source, and, on the other hand, the sink and the elements connecting it to the device. These elements allow for modifying at will the boundary conditions that condition the coupling of the device with the source and the sink, which is a central question for the optimization. Among the latter, we can distinguish systems whose intrinsic parameters are constant, as is the case for most machines, and systems, whose intrinsic parameters are subject to modification, as is the case for living or societal systems. These latter are subject to potential developments and evolution, which are not possible for the above-mentioned machines. By potential development, we consider the case of living systems, societies or organisms, which can, under conditions of energy and matter supply, develop, maintain, or regress.

In the case of systems under Neumann boundary conditions, the system is somehow fed by a constant current of energy and/or matter, which guarantees the maintenance of the system as much as it constrains its development. Under such conditions, the possible development of the system is limited by the value of the current of matter and/or energy. In the case of Dirichlet systems, there are no restrictions on access to the resource, except for the intrinsic limitations of the conversion device. As a result, the currents of energy and matter may diverge completely, if the characteristics of the device lend themselves to it. The same reasoning applies to the production and rejection of waste to the sink. Access to the resource and waste production are therefore both dependent on these boundary conditions. Let us consider, as an historical illustration, the situation of the industrial revolution, which saw the rise of the use of fossil energy [12]. The latter are by definition stock resources that lead the human societies to find themselves in Dirichlet conditions, as far as access to the resource is concerned. Concerning the waste rejected to the sink, the Dirichlet’s condition has been the norm, as long as the planet has been considered a bottomless sink. On the other hand, if we consider the situation before the industrial revolution, it can be noted that the main resource for development, which is the food resource, was dependent on Neumann-type boundary conditions, due to the subjection to solar flux. Without going further into this illustration, which is beyond the scope of this article, we can nevertheless observe the importance of boundary conditions, both on the functioning of systems, but also for their possible evolutions. Indeed, in the case of boundary conditions of the Neumann type, there is no possibility of development, in the sense of increasing the current of energy and matter that feed the conversion device. Consequently, there is no possibility of any increase of the quantities. On the other hand, there are possibilities of increase of the quality because the conditions of coupling between energy and matter may change, as the history of life proved it.

On the other hand, in the case of Dirichlet boundary conditions, there is no limit to the increase in energy and matter currents, which could lead to their possible divergence. It should be noted that the actual Dirichlet conditions for the access to the energy for the human species are quite singular in the history of the living systems. In order to remain explicit and relatively simple to address, these questions need to be modeled in the most compact form possible. This why we propose to describe a generic thermodynamic machine in order to guarantee a general character to the developments of this article. Many extensions and refinements can be added, as for previous systems in the literature [6,10].

### 2.2. Thermodynamic Device

The proposed thermodynamic system is described in Figure 1. It consists of a reservoir providing the resource and a sink receiving the waste, with the respective potentials $\Pi_{1}^{R}$ and $\Pi_{1}^{S}$ fixed at constant values. Between these two reservoirs is the energy conversion device which is the place of coupling between a current of matter $I_{2}$, and a current of energy $I_{E}$. The energy current entering the system is associated with an incoming entropy current, $I_{1}$, with $\Pi_{1}$ its conjugated potential. In the case of a thermal system of heat current $I_{Q}$, temperature *T* and entropy current $I_{S}$, we would simply have $\Pi_{1}I_{1}=I_{Q}=TI_{S}$ so $I_{1}$ would be the classical entropy current. The current of matter is defined by $I_{2}$ and its conjugated potential $\Pi_{2}$. The energy currents budget finally writes $I_{E}=\Pi_{1}I_{1}+\Pi_{2}I_{2}$. We recognize the fractions of dispersed energy, $\Pi_{1}I_{1}$, and concentrated energy, $\Pi_{2}I_{2}$, which are a generalization of the notions of heat and work extended to the case of non-thermal systems [13,14]. The coupling term between energy and matter is defined, under $I_{2}=0$ condition, as $\alpha =-{\left(\delta \Pi_{2}/\delta \Pi_{1}\right)}_{I_{2}}$. The geometry of the system is given by its length *L* and its cross-section *A*. The two currents of energy and matter are then associated with two conductivities $\sigma_{1}$ and $\sigma_{2}$, which, at the integrated scale, behave like two resistive dipoles $R_{1/2}=\frac{1}{\sigma_{1/2}}\frac{L}{A}$. The connection of the conversion zone with the two reservoirs is defined by the coupling resistors $R_{+}$ and $R_{-}$, which allow the boundary conditions to be set, at will, between Dirichlet conditions $\left(R_{+}=R_{-}=0\right)$, or Neumann conditions, where $R_{+}$ and $R_{-}$ diverge. This type of configuration is not in itself new, and has already been used in specific systems [14,15]. In particular, it has been shown that, under these conditions, the way the system operates is partially governed by the feedback effects induced by boundary conditions. Some of this feedback can lead to the presence of oscillations. It should be noted that these processes do not violate the first principle in that they are not self-sustained oscillations, at least from an energy point of view. They do not violate the second principle either, since these structures are highly dissipative and are only maintained by a continuous supply of energy. It can also be noted that the incoming current of energy is used to produce a potential difference, which, if maintained, allows the circulation of the matter under the action of the thermodynamic force, which is defined from the gradient of the potential. This type of analysis of thermodynamic conversion has been used with success by Alicki in various systems [16,17]. This description of two coupled currents can, of course, be extended to a larger number of coupled currents without changing the spirit of the study.

As it is represented, the system is therefore quite generic. The main determinants of functioning are thus summarized by three terms, the capture of the resource, its conversion into a usable form, and the rejection of waste. It is clear that ideally the target is the one where the output power would be maximum and the amount of energy released would be minimal. The study of the limits to achieving this target is one of the objectives of this article. As the coupling parameter for the conversion, the α parameter is therefore central since it determines the system’s ability to convert energy into a usable form. A naive picture may suggest that the largest possible α value necessarily leads to the most efficient system, but this is not correct, as we will see now.

## 3. Local Energy Conversion

### 3.1. Presentation

At the local level, energy conversion is produced by coupling the energy and matter currents flowing through the device. These currents are generated by the presence of differences between the two thermodynamic potentials $\Pi_{1}$ and $\Pi_{2}$. This local modeling is therefore based on the three parameters of conductivity associated with energy transport, $\sigma_{1}$, conductivity associated with matter transport, $\sigma_{2}$, and the coupling coefficient between the gradients of the two potentials, α. We deduce from this the formulation of local Onsager matrix, where $\nabla =\frac{d}{dx}$ is the spatial gradient, here reduced to 1D in order to simplify the description.
(1)$$\left(\begin{array}{c} J_{2}\\ J_{E} \end{array}\right)=\left(\begin{array}{cc} L_{11} & L_{12}\\ L_{21} & L_{22} \end{array}\right)\left(\begin{array}{c} -\nabla \left(\frac{\Pi_{2}}{\Pi_{1}}\right)\\ \nabla (\frac{1}{\Pi_{1}}) \end{array}\right).$$
$J_{E}$ and $J_{2}$ are the densities of the two currents, and are extensive and conservative quantities. Given the differential form $J_{E}=\Pi_{1}J_{1}+\Pi_{2}J_{2}$, the equality of non-diagonal terms $L_{12}=L_{21}$ is insured according to the choice of the correct potentials $-\frac{\Pi_{2}}{\Pi_{1}}$ and $\frac{1}{\Pi_{1}}$ [18,19]. The four terms of the matrix are therefore reduced to three, $\sigma_{1}$, $\sigma_{2}$ and α, whose correspondences with the coefficients $L_{ij}$ are
(2)$$\begin{array}{cc} \sigma_{1} & =\frac{1}{\Pi_{1}^{2}}\left[\frac{L_{11}L_{22}-L_{21}L_{12}}{L_{11}}\right], \end{array}$$
(3)$$\begin{array}{cc} \sigma_{2} & =\frac{L_{11}}{\Pi_{2}}, \end{array}$$
(4)$$\begin{array}{cc} \alpha & =-\frac{\Delta \Pi_{2}}{\Delta \Pi_{1}}=\frac{1}{\Pi_{1}}\frac{L_{12}}{L_{11}}. \end{array}$$


In the absence of a matter gradient, the energy conductivity can be defined as $\sigma_{\Pi_{2}}=\sigma_{1}\left[1+\alpha^{2}\sigma_{2}/\sigma_{1}\Pi_{2}\right]$. The figure of merit is then defined as
(5)$$F_{m}=\frac{\alpha^{2}R_{1}}{R_{2}}\Pi_{2}=\frac{L_{12}^{2}}{L_{11}L_{22}-L_{21}L_{12}}.$$


It is known that the ratio $\sigma_{2}/\sigma_{1}$, therefore $F_{m}$, is a direct measure of the intrinsic capacity of energy conversion. $F_{m}$ can be related to the ratio of the equivalent specific heats by the expression $\gamma =\frac{C_{\Pi_{2}}}{C_{I_{2}}}=F_{m}+1$. In their seminal paper, Kedem and Caplan derived the following expression of the coupling parameter between the two fluxes involved in the conversion process [13]:
(6)$$q=\frac{L_{12}}{\sqrt{L_{11}L_{22}}}=\sqrt{\frac{F_{m}}{1+F_{m}}}$$
an expression that explicitly includes the kinetic coefficients $L_{ij}$. The figure of merit and the coupling factor *q* are equivalent in terms of measure of the system performance: the larger their (absolute) values, the better the energy conversion system. This can be evidenced by the derivation of the local maximal efficiency of the conversion process in generator mode, $\eta_{\max}$:
(7)$$\eta_{\max}={\left(\frac{1+\sqrt{1-q^{2}}}{q}\right)}^{2}=\frac{\sqrt{\gamma }-1}{\sqrt{\gamma }+1}.$$


### 3.2. Entropy Production and Efficiency

The volumetric entropy production rate is given by the summation of the force-flow products,
(8)$$\dot{S}=J_{2}\nabla \left(-\frac{\Pi_{2}}{\Pi_{1}}\right)+J_{E}\nabla \left(\frac{1}{\Pi_{1}}\right)=-\frac{1}{\Pi_{1}}\left[J_{2}\nabla \Pi_{2}+J_{1}\nabla \Pi_{1}\right].$$


In the case of a reversible process $\dot{S}=0\;$ so does $J_{2}\nabla \Pi_{2}+J_{1}\nabla \Pi_{1}$. We get $-\frac{J_{2}\nabla \Pi_{2}}{\Pi_{1}J_{1}}=\frac{\nabla \Pi_{1}}{\Pi_{1}}=\eta_{C}$, where $\eta_{C}$ is the Carnot efficiency. This leads to the general expression of the local efficiency,
(9)$$\eta =-\frac{J_{2}\nabla \Pi_{2}}{J_{1}\Pi_{1}}<\eta_{C}.$$


Let us define the reduced current as
(10)$$j=\frac{\alpha J_{2}}{J_{1}},$$
which is the ratio between the entropy carried by the transport of the matter, divided by the total entropy transported. In the case of a reversible process, both terms are equal so j=1 [20]. This expression shows three regions for the η(j) meaning. For 0<j<1, the device works as a generator. For j<0 and j>1, the device works as a receptor. For reasons of brevity, we will mainly deal with the generator configuration in this article.

Rewriting the Onsager matrix in more suitable form [21], we get
(11)$$\left(\begin{array}{c} J_{2}\\ \Pi_{1}J_{1} \end{array}\right)=\left(\begin{array}{cc} \sigma_{2} & \alpha \sigma_{2}\\ \alpha \Pi_{1}\sigma_{2} & \gamma \sigma_{1} \end{array}\right)\left(\begin{array}{c} -\nabla \Pi_{2}\\ -\nabla \Pi_{1} \end{array}\right).$$


Then,
(12)$$j=\frac{\eta \alpha \Pi_{1}\sigma_{2}-j\alpha \sigma_{2}\nabla \Pi_{1}}{\eta \alpha \Pi_{1}\sigma_{2}-\frac{j\gamma \sigma_{1}}{\alpha }\frac{\nabla \Pi_{1}}{\Pi_{1}}}.$$


Thus,
(13)$$\eta =\eta_{C}j\frac{j\gamma -\frac{\alpha^{2}\sigma_{2}}{\sigma_{1}}\Pi_{1}}{j\frac{\alpha^{2}\sigma_{2}}{\sigma_{1}}\Pi_{1}-\frac{\alpha^{2}\sigma_{2}}{\sigma_{1}}\Pi_{1}},$$
where $\gamma =\frac{\alpha^{2}\sigma_{2}}{\sigma_{1}}\Pi_{1}+1$. After a few algebra, we get
(14)$$\eta =\frac{\eta_{C}}{\left(\gamma -1\right)}\frac{\gamma j^{2}-\left(\gamma -1\right)j}{j-1}$$
η presents a maximum for $j_{opt}=1+\sqrt{\frac{1}{\gamma }}$ for a receptor mode, and $j_{opt}=1-\sqrt{\frac{1}{\gamma }}$ for a generator mode. Both optima reduce to j=1 in the ideal case, when γ diverges, where we recover the Carnot efficiency. In this diverging case, the system do not present anymore dissipation production, and the equivalence between the receptor and generator modes is a proof of the absence of causality of the Carnot configuration. This absence of causality is another name for reversibility. We then recover the Kedem–Caplan expression of the maximal efficiency, $\eta_{\max}=\eta_{C}\frac{\sqrt{\gamma }-1}{\sqrt{\gamma }+1}$ for the generator mode, and $\eta_{\max}=\eta_{C}\frac{\sqrt{\gamma }+1}{\sqrt{\gamma }-1}$ for the receptor mode. Let us now plot the efficiency versus the reduced current, as reported in Figure 2.

As expected for the maximum performance achieved, $\eta_{max}$ is an increasing function of the figure of merit. On the other hand, it also appears that the sensitivity to fluctuations in *j* becomes all the more important as $\eta_{max}$ is important. This is confirmed by estimating the value of the slope in the vicinity of the maximum yield, which is $\partial \eta /\partial \left(j\right)\approx -2\eta_{\max}F_{m}$. The larger the figure of merit, the steeper the slope. This local description allows us to conclude that the performance of the device is obtained at the cost of a constraint of stability of the operating points, directly driven by the value of the figure of merit. As an intrinsic quantity, the figure of merit defines the performance ceiling beyond which it cannot be exceeded. It is clear from the figure that the system defined by a high figure of merit exceeds in performance all the systems of lower figure of merit value. However, this result is strongly weighted by the fact that, for excursions of *j* around the optimal value, the efficiency falls rapidly. Then, it is not necessarily relevant to look for a device with a large figure of merit, without first inventorying the operating range that will be brought to run this device. For simplicity’s sake, we have only dealt here with the case where the system works as a generator, which is obtained by 0<j<1. It is clear that the same study can be carried out for the case where the system operates as a receptor, instead of working as a generator. This situation, well known for thermal machines, corresponds to heat pump operation. More broadly, and in the case of non-thermal machines, this case actually corresponds to the operation in recycling mode where the treated quantity undergoes regeneration. It should be noted that the expression of performance refers only to γ, and therefore to the figure of merit, without specifying any contribution from $\sigma_{1}$, $\sigma_{2}$ and α, respectively. The local level is totally blind to these issues so we now consider the situation of the entire system to see the relative contributions.

## 4. Global Conversion System

### 4.1. Presentation

In accordance with the diagram in Figure 1, the device of the conversion zone is connected to its reservoirs via the two resistors $R_{+}$ and $R_{-}$, which makes it possible to explore all boundary conditions. The presence of $R_{+}$ and $R_{-}$ may lead to the pinching of the potential difference $\Pi_{1}^{+}-\Pi_{1}^{-}$ according to the system operating point. More precisely, $R_{+}$ governs the limitation of access to the resource while $R_{-}$ reflects possible saturation effects of waste disposal. This global model, although limited, makes it possible to approach the behavior of many systems, including living systems, depending on whether the resource is abundant or scarce, and whether waste disposal, including thermal waste, is easy or not. Living system and non-living systems differ from the fact that the energy current is never zero in living systems, so $R_{1}$ is always finite, and there is a non-zero resting point. On the contrary, a non-living system may have a zero resting point, with zero energy current, so $R_{1}$ may be infinite in these systems. Let us consider the set of the four equations that governs the functioning of the system (see Appendix A):
(15)$$\begin{array}{cc} I_{E-} & =\alpha \Pi_{1}^{-}I_{2}+\left(1-\varphi \right)R_{2}I_{2}^{2}+\frac{\left(\Pi_{1}^{+}-\Pi_{1}^{-}\right)}{R_{1}}, \end{array}$$
(16)$$\begin{array}{cc} I_{E-} & =\frac{\left(\Pi_{1}^{-}-\Pi_{1}^{S}\right)}{R_{-}}, \end{array}$$
(17)$$\begin{array}{cc} I_{E+} & =\alpha \Pi_{1}^{+}I_{2}-\varphi R_{2}I_{2}^{2}+\frac{\left(\Pi_{1}^{+}-\Pi_{1}^{-}\right)}{R_{1}}, \end{array}$$
(18)$$\begin{array}{cc} I_{E+} & =\frac{\left(\Pi_{1}^{R}-\Pi_{1}^{+}\right)}{R_{+}}. \end{array}$$


These equations have their origin in the integration of the local form described in the previous paragraph. These developments have been the subject of previous articles [14,22], and will not be re-described here. φ controls the dissipation fraction that returned to the source or to the sink. In the following, we will choose φ=0. This choice is not critical here since the effect of φ=0 is driven by $R_{2}$, which is equal to zero.

### 4.2. Devices with Zero Resting Point

First of all, we consider that $R_{2}=0$ and $R_{1}$ diverge, in order to separate the contributions of entropy production and internal dissipation. $R_{2}$ governs the current of matter, so we therefore consider that this current may not be limited, so there is no intrinsic dissipation within the device. The figure of merit of the device is then infinite and we may expect to reach the ideal conditions and the Carnot efficiency. However, the classical discussion around the Carnot efficiency is based on pure Dirichlet boundary conditions, which is clearly not the case here, so we have to consider the new conditions introduced by the modification of the boundary conditions. In the present configuration of zero resting point systems, the general equations (see Appendix A) can be summarized as
(19)$$\begin{array}{cc} I_{E-} & =\frac{\Pi_{1}^{S}I_{2}}{\frac{1}{\alpha }-R_{-}I_{2}}, \end{array}$$
(20)$$\begin{array}{cc} I_{E+} & =\frac{\Pi_{1}^{R}I_{2}}{R_{+}I_{2}+\frac{1}{\alpha }}, \end{array}$$
with the output power given by $P=I_{E+}-I_{E-}$.

The plots in Figure 3 summarize the behavior of the global system. The output power presents a maximum and two zero values. The first value corresponds to the case where the efficiency reaches its maximum. This situation is obtained for $I_{2}=0$, so $I_{E-}=I_{E+}=P=0$. This means that no matter or energy can flow through the system, which is a totally useless situation for a physical system. The second zero power value is reached for a current of matter $I_{2_{sc}}$, named the short-circuit current, by analogy with electronics. In this situation, the produced power is completely re-dissipated inside the system. $I_{2_{sc}}$ is therefore an ultimate operating point for the system, working as an energy generator. For a truly efficient operation, it is therefore necessary to try to push $I_{2_{sc}}$ to large values, which are obtained by getting as close as possible to Dirichlet conditions. In the general case, the approximate expression of this current is
(21)$$I_{2sc}\approx \frac{1}{\alpha }\frac{\Delta \Pi_{1}}{R_{-}\Pi_{1}^{S}+R_{+}\Pi_{1}^{R}},$$
which confirms that Dirichlet’s conditions where $R_{-}=R_{+}\approx 0$ are to be sought, if accessible. Since the resting point here is zero, the power curve necessarily intercepts that of $I_{E-}$. Beyond this interception point, the system is in a situation where it releases more waste than it produces output power. We call *critical point* the point where $P=I_{E-}$, reached for $I_{2cp}$. The fact that power is not a monotonous function of $I_{2}$ is actually quite unexpected because, to the extent that $R_{2}=0$, the total absence of intrinsic viscosity should not lead to any limit to $I_{2}$. However, if we carry out a development at the first order of the expression of power we find
(22)$$P\approx \left[\alpha \Delta \Pi_{1}-\left(\Pi_{1}^{R}R_{+}+\Pi_{1}^{S}R_{-}\right)\alpha^{2}I_{2}\right]I_{2}$$
which clearly indicates the presence of a viscous friction term $R_{fb}$,
(23)$$R_{fb}\approx \alpha^{2}\left(\Pi_{1}^{R}R_{+}+\Pi_{1}^{S}R_{-}\right)$$
which reduces the transport of the matter, even though the intrinsic viscosity, i.e., $\frac{1}{\sigma_{2}}$, associated with the transport of the matter, is zero. This additional dissipation is a pure feedback effect that is due to the presence of boundary conditions at the general limits where $R_{+}$et $R_{-}$ are non-zero. This additional dissipation can only be rendered null if $R_{+}=R_{-}=0$, i.e., a strict Dirichlet condition, which is, in reality, only very rarely observed. Note that the condition α=0 leads to the same result but it is useless because in this case the transport of energy and matter are fully decoupled, and the device does not convert the energy anymore. The conditions $R_{2}=0$ and $R_{1}\to \infty$ determine the performance envelope for a system with an ideal conversion zone. In particular, it is noted that, although $I_{E+}$ and $I_{E-}$ are increasing functions of the current of matter $I_{2}$, the growth rate of the energy waste current $I_{E-}$ always ends up reaching that of the energy current $I_{E+}$ supplied to the system. In addition, even in the case of a system whose core is composed of an ideal device, $\left(R_{2}=0,\;R_{1}\to \infty \right)$, the increase in the current of matter inexorably leads to an increase in the current of waste in larger proportions to the rate of supply of resources. The only way out is to limit the current of matter to values below a threshold, which may be that of maximum power, maximum efficiency, minimum waste generation, or below the critical point. In the Figure 3, the response is given for two different values of the coupling parameter α.The influence of α is quite surprising. At first we observe that the lower is α and the lower are the output power and efficiencies, as expected for a lower conversion level of the energy. However, in the same time, the short-circuit current is strongly enhanced, opening the way to a large range of $I_{2}$ working points for the transport of the matter. This is due to the $\alpha^{-2}$ dependency of $I_{2sc}$. This leads to the conclusion that *the search for a very efficient system is in contradiction with the search for a very adaptable system.*

Let us now focus on the issue of the trade-off between power efficiency and waste generation. The Figure 3a represents the curves of the production efficiency $\eta_{prod}=P/I_{E+}$ and the waste efficiency $\eta_{waste}=P/I_{E-}$. Note that $\eta_{prod}$, which is the traditional efficiency, is limited by the Carnot efficiency but $\eta_{waste}$ is not, since it does not refer to the traditional expression of efficiency but is just an extension of the notations. $\eta_{prod}$ is bounded by a zero value, which corresponds to zero power, and a maximum efficiency point, reported in Figure 3b. Between these two values, the system presents a maximum of the power, which absolutely does not coincide with the maximum efficiency. In this configuration the MPP or mEP operations are clearly disjointed as already mentioned [23,24]. Let us now consider the cost of carrying out a unitary process. By unitary process we consider a process standardized by the value of the associated transport of matter, i.e., the ratio between the energy currents and the matter current. We call this quantity Cost Of Energy, i.e., COE. This makes it possible to consider energy expenditures with regard to the associated matter transformation along a unitary displacements. In other words, COE can measure the amount of energy needed to be rejected as a waste, for displacing the matter from a unit length. This quantity is already known in biology as Cost Of Oxygen Transport (COT), where it has made it possible to qualify a unit displacement with regard to the energy released in the form of waste [25,26]. Here, we extend the notion in a more general form where COE is defined by $COE_{+}$ which is the cost of energy needed to feed the system, and $COE_{-}$ which is the cost of waste energy that is rejected, so,
(24)$$COE_{+/-}=\frac{I_{E+/-}}{I_{2}}$$


Note that the $COE_{+}$ is a strictly decreasing function of $I_{2}$ and $COE_{-}$ is a strictly increasing function of $I_{2}$. This means that the cost of energy needed for a unitary process decrease when $I_{2}$ increases but, in the same time the amount of waste always increases. There is therefore no optimum to consider any minimization of the waste. In addition, it is important to note that the $R_{1}^{-1}=0$ configuration is the only one that provides the strong coupling conditions, for which the energy and matter currents are roughly proportional [10]. In this case, the Onsager matrix has a zero determinant. This situation is an idealization of the transport of energy entirely achieved by the transport of matter. In other words, it is a question of considering that the behavior of out-of-equilibrium thermodynamics may be equivalently described by pure mechanics. This is obviously never fully encountered unless it is considered that a $\Delta \Pi_{1}$ difference can persist without an associated current of matter existing. This is the purpose of the following paragraph.

### 4.3. Devices with Non Zero Resting Point

The study of devices with non zero resting points concern the case of all systems for which a shutdown means death. Indeed, unlike a machine, all living systems are never totally shut down, and always keep a minimum operating point value, which we call basal, also known as a resting point. This situation corresponds to the case where $R_{1}$ has a finite value. While remaining, for the moment in the case where $R_{2}=0$, we can develop the main results from this configuration. The general equations of the system are given in Appendix B. In this situation, the efficiency, nor the power, can reach the previous values, as reported in the Figure 4. At the resting point $I_{2}=0$, the system is in its basal configuration where P=0, so $I_{E+}=I_{E-}=B$ with,
(25)$$B=\frac{\Delta \Pi_{1}}{R_{+}+R_{1}+R_{-}}$$


The typical response of systems with non zero resting points is given in the Figure 4. One can notice that the general shape is not strongly modified from the case of zero resting point configurations, except the presence of a non zero current of energy even at zero $I_{2}$ and a slight modification of the short-circuit point. Regardless of the reduction in efficiency introduced by the presence of $R_{1}$, the search for a system with a very low basal point requires to be located in a configuration close to Neumann conditions where $R_{+}$ and $R_{-}$ have very large values. This is not problematic except that it requires the system to operate at low values of $I_{2}$, in order to limit the dissipation due to the term $R_{fb}$. There is therefore a fundamental contradiction between having a system with low resting power consumption and a system that can provide significant power. It is clear that a sober system, in the sense of its consumption at rest, is unsuited to the production of significant power, without leading to significant dissipation at high speed, or equivalently, high $I_{2}$. If such a power is sought, then it implies that the boundary conditions should be of Dirichlet like with $R_{+}\approx R_{-}\approx 0$. However, in this case the system will have a necessarily high rest consumption. Compared to systems with a zero resting point, it can be seen that the maximum power operating point and maximum efficiency operating point tend to approach each other as $R_{1}$ increases. In this configuration, as can be derivated in [27], the feedback resistance is approximately given by
(26)$$R_{fb}\approx \frac{\alpha^{2}\langle \Pi_{1}\rangle }{\frac{1}{R_{+}+R_{-}}+\frac{1}{R_{1}}}=R^{*}\alpha^{2}\langle \Pi_{1}\rangle$$
where $R^{*}=\frac{\left(R_{+}+R_{-}\right)R_{1}}{R_{1}+R_{+}+R_{-}}$ and $\langle \Pi_{1}\rangle =\Pi_{1}^{R}/2+\Pi_{1}^{S}/2$.

Compared to the previous configuration the dissipation introduced by the presence of $R_{fb}$ can now be modified whatever are the boundary conditions because $R^{*}<Min\left(R_{+}+R_{-},R_{1}\right)$. More precisely, in the case of Neumann-like boundary conditions, there is a restriction to the value of $R_{1}$ where $R_{1}\ll R_{+}+R_{-}$ is expected. Under Dirichlet-like boundary conditions $R_{+}$ and $R_{-}$ are small so there is no condition on $R_{1}$. Consequently, a system with a very low basal point, with large values of both $\left(R_{+},\;R_{-}\right)$ (Neumann like) and $R_{1}$ will suffer from a large $R_{fb}$ and is then limited to very low $I_{2}$ currents. If the boundary conditions are more like Dirichlet conditions, then $R_{fb}$ keeps low but the low basal level now imposes that $R_{1}$ strongly increases, which reduced both the available power *P* and the efficiency. Thus, we can see that there is no room for a powerful and efficient system working in all conditions. The main trade-off is between power and efficiency, but it ultimately extends beyond that.

From a rather general point of view, the incoming energy current $I_{E+}$ makes it possible to establish and maintain, thanks to the presence of $R_{1}$, a potential difference that permits the production of output work. On this point, we join the work of Alicki [16], who considers that the incoming energy current makes it possible to maintain a difference in potential, exactly as a pump would do. This situation is particularly described in the case of photovoltaic structures, with a difference in electrochemical potential [16], or in the case of muscles where the attachment and release cycles of actin and myosin structures lead to the maintenance of a force [28]. It should be noted that, depending on the position of the resting point, the power curve can intercept between zero and twice the $I_{E+}$ curve. It can therefore be seen that, in the case of systems with a relatively low resting point, there may be an area for which the power produced is greater than the power released as a waste. More intriguing, this area can start with a non-zero value of $I_{2}$. In other words, there may be systems for which the situation $I_{2}\neq 0$ leads to a proportionally smaller waste production than at rest. Systems with a non-zero resting point therefore present very different optima than non-living systems, whose zero resting point leads to minimizing power by stopping the machine. By using the definition $COE_{-}=I_{E-}/I_{2}$, we can plot its response according to $I_{2}$. It should be noted that the COE− has a minimum value, which does not coincide with the maximum power point. This defines a new operating point for the system, which characterizes the situation where the system minimizes its production of waste.

An illustration of this can be given if we consider the motion of living systems. Let us consider that the task to be accomplished consists in moving the body over a unit distance, the question arises as to how fast this operation will lead to a minimum of waste, essentially in the form of heat and metabolic degradation products. It is clear that displacement here corresponds to the transport of matter, and is therefore assimilated to $I_{2}$ proportional to the speed of travel as previously said. There is an abundant amount of literature showing that there exists a minimum of the so-called $COT\equiv COE_{-}$ point for all animals for which movement appears to be favored when the COT is minimal [25,26]. As expected, see Figure 4, $COE_{-}$ and COE+ curves have a common point at the short circuit point. We previously saw that Dirichlet’s conditions, $R_{+}=R_{-}=0$, were those that minimized the feedback resistance $R_{fb}$ and allowed for considering potentially a divergence of the current of matter and the output power. This simple observation shows that strict Dirichlet’s conditions are simply nonphysical. Nevertheless, one can consider that this condition can be approached. However, the presence of $R_{+}$, $R_{-}$ and $R_{1}$ in series shows that Dirichlet’s condition is asymptotically obtained only if the ratios $R_{+}/R_{1}$ and $R_{-}/R_{1}$ are negligible, which imposes an important value for $R_{1}$, and therefore a high value of the basal power. *We therefore see the emergence of a paradox, which, seeking to minimize the dissipation due to $R_{fb}$ leads to the constraint of high consumption at rest. The same system cannot therefore be both very powerful and very energy-efficient at its resting point.* We find here the generalization of a well-known situation, for example for the thermal engines of vehicles, in which the engine’s displacement determines its ability to produce power, as well as its efficiency.

### 4.4. Internal Dissipation Devices

Let us now consider the introduction of the dissipative term $R_{2}$. The output power of the system is now represented by Figure 5. As a thermodynamic engine, the system provides a power $P=\alpha \left(\Pi_{1}^{+}-\Pi_{1}^{-}\right)I_{2}$ as already defined. The efficiency of this part of the system is given by $\eta_{2}=\frac{P-R_{2}I_{2}^{2}}{P}$. Thus, the total efficiency of the system is
(27)$$\eta_{sys}=\eta_{1}\eta_{2},$$
with $\eta_{1}=\frac{P}{I_{E_{+}}}$. Compared to the previous configurations, both the power, the short-circuit current $I_{sc}$, and the efficiency are now reduced. The influence of $R_{2}$ appears to be always detrimental, which was not the case for $R_{1}$. It is clear that one should look for minimal $R_{2}$ if possible. In other words, in the expression of the figure of merit, there is a constraint on $R_{2}$. At first, both α and $R_{1}$ seem to be non-constrained, and the same figure of merit can be obtained for various values of the couple $\left(\alpha ,\;R_{1}\right)$. Nevertheless, as we have mentioned, the present description shows that $R_{2}$ is linked in series with $R_{fb}$. Consequently, the constraint on $R_{2}$ can be relaxed to the condition $R_{2}\ll R_{fb}$. According to the expression $R_{fb}\approx \frac{\alpha^{2}\langle \Pi_{1}\rangle }{\frac{1}{R_{+}+R_{-}}+\frac{1}{R_{1}}}$, this leads to the condition $1+\frac{R_{1}}{R_{\Sigma }}<\frac{\alpha^{2}R_{1}}{R_{2}}\langle \Pi_{1}\rangle$ where we recognize the figure of merit, so the condition becomes
(28)$$1+\frac{R_{1}}{R_{\Sigma }}<F_{m},$$
where $R_{\Sigma }=R_{+}+R_{-}$. According to the previous observation, the minimization of the dissipation occurring from the $R_{fb}$ term imposes that $\frac{R_{1}}{R_{\Sigma }}$ should be large enough. Thus, we now get a supplementary condition for $F_{m}$. In this expression, the boundary conditions and the intrinsic performances of the device are considered together. Under Dirichlet conditions, $1+\frac{R_{1}}{R_{\Sigma }}$ diverges so the system keeps its level of dissipation low only in the case of a very large figure of merit, and is forced to work at very low $I_{2}$ values. Under Neumann conditions, $R_{\Sigma }$ diverges and then the condition on the figure of merit is then relaxed. Ideally, even when achieved asymptotically, one might want to achieve maximum power, as well as minimal waste production, combined with maximum efficiency. *We conclude that looking for maximum efficiency always leads to approaching the Carnot point, which is, even in an out-of-equilibrium description, the point where power production is canceled out.*

## 5. Entropic Point of View

The previous power budget analysis highlighted three classes of systems: systems with a zero resting point, systems with a non-zero resting point, and, finally, systems with an additional internal dissipation term $R_{2}$. Let us consider these three classes again from the entropic point of view.

### 5.1. Devices with Zero Resting Point

The production of entropy from the presence of $R_{-}$ and $R_{+}$ is given respectively on both sides of the device by
(29)$$\begin{array}{cc} {\dot{S}}_{E+}= & I_{E+}\left(\frac{1}{\Pi_{1}^{+}}-\frac{1}{\Pi_{1}^{R}}\right)=\frac{\alpha^{2}I_{2}^{2}R_{+}}{1+\alpha I_{2}R_{+}}, \end{array}$$
(30)$$\begin{array}{cc} {\dot{S}}_{E-}= & I_{E-}\left(\frac{1}{\Pi_{1}^{S}}-\frac{1}{\Pi_{1}^{-}}\right)=\frac{\alpha^{2}I_{2}^{2}R_{-}}{1-\alpha I_{2}R_{-}}. \end{array}$$
The results are given in Figure 6.

There is clearly an asymmetry in the two entropy productions. Indeed, if the two contributions initially increase in a quadratic form with the current of matter, the contribution of the resource side, ${\dot{S}}_{E+}$, tends to a linear progression independent of the coupling condition $R_{+}$, while the contribution on the waste rejection side ${\dot{S}}_{E-}$ tends to diverge as soon as $I_{2}\approx 1/\alpha R_{-}$. It is surprising to see that, in addition, this divergence is more marked as the coupling factor α between energy and matter is important. *There is therefore no other solution than to make $R_{-}$ as small as possible, and therefore reject all the waste easily.* This is an additional constraint for the design of efficient systems.

### 5.2. Devices with Non-Zero Resting Points

Let us now look at the configuration of non-zero resting point systems, while keeping $R_{2}\approx 0$. In this case, the general expressions become
(31)$$\begin{array}{cc} {\dot{S}}_{E+}= & I_{E+}\left(\frac{1}{\Pi_{1}^{+}}-\frac{1}{\Pi_{1}^{R}}\right)=\frac{R_{+}I_{E+}^{2}}{\left(\Pi_{1}^{R}-R_{+}I_{E+}\right)\Pi_{1}^{R}}, \end{array}$$
(32)$$\begin{array}{cc} {\dot{S}}_{E-}= & I_{E-}\left(\frac{1}{\Pi_{1}^{S}}-\frac{1}{\Pi_{1}^{-}}\right)=\frac{R_{-}I_{E-}^{2}}{\left(R_{-}I_{E-}+\Pi_{1}^{S}\right)\Pi_{1}^{S}}. \end{array}$$


The results are given in Figure 7 where $I_{E+}$ and $I_{E-}$ are defined according to the Appendix B. We can see that the presence of $R_{1}$ reintroduces a significant symmetry between the two contributions to the entropy production. Moreover, the question of the importance of the quality of the coupling on the resource side, by minimizing $R_{+}$, or to the rejection side, by minimizing $R_{-}$, is now of equal importance.

### 5.3. Internal Dissipation Devices

For internally dissipated devices, the term $R_{2}$ produces a quadratic dissipation $R_{2}I_{2}^{2}$. We have seen before that the presence of $R_{2}$ never brings any advantage in terms of energy conversion since it only contributes to lowering the power available at the output of the system. As this dissipation diffuses into the system, it is itself a source of entropy, as shown in Figure 8. At this stage, it is important to know how this dissipation occurs. In the case of some thermal systems, an analytical calculation can be carried out that leads to an equal distribution of this dissipation between the resource and the sink, i.e., φ=0.5, see Appendix B in accordance with [22]. In other systems, such as muscles subjected to moderate stress, this dissipation is considered to be completely rejected into the sink (φ=0) [14]. For some living systems, including homeothermic species, it is likely that a fraction of this dissipation is partially released, and partially used to maintain the central temperature of the body, leading to a value φ≈1, depending on outdoor conditions. One example is the case of vaso-dilatation and vasoconstriction of peripheral vessels, which is a solution for modulating the value of $R_{-}$ and consequently reject less, or more, heat outside of the body.

## 6. Adaptable or Adapted?

The study of the behavior of a generic system composed of a conversion device, and the boundary conditions to the reservoirs, now allows us to establish several observations. First, the search for the best device, in terms of power and efficiency, can be summarized by the search for the largest figure of merit $F_{m}$. However, this result must be modulated by the fact that the value of $F_{m}$ is determined by the set of the three parameters $R_{1}$, $R_{2}$, and α which, at this stage, do not present any constraints. In addition, few thermodynamic devices have a single operating point, but are generally expected to operate over a wide range of uses that principally means large range of $I_{2}$. In the precedent paragraph, we concluded that the greater the figure of merit, the smaller the effective operating range becomes. Indeed, for such a narrow range, the users must then conform quite strictly to that imposed by the value of the figure of merit of the device. This observation explains quite simply why the consumption observed by vehicle drivers is always larger than that reported by vehicle manufacturers, since the actual conditions of use never coincide with the test conditions. Similarly, the measured performance of equipment in dwellings, as well as the performance of the dwellings themselves, is below the expected performance during construction. This observation leads to the recommendation that devices intended to operate over a wide range of uses should not be designed solely on the basis of their maximum performance in terms of efficiency and power. Beyond this observation, the question arises of determining, within a system, which of the three parameters $R_{1}$, $R_{2}$, and α should be optimized as a priority. We can first conclude that, unless there are situations where dissipation is explicitly sought, $R_{2}$ must be systematically minimized. With regard to $R_{1}$, we have seen that its choice determines two categories of systems, depending on whether $R_{1}$ is zero or not. It must be noticed that $R_{1}=0$ is not possible for living systems because a resting point does exist until the death. In the category where $R_{1}=0$, the operating range of the system is limited by the feedback effects that introduce an excess dissipation term $R_{fb}$. Note that this term can be minimized if the boundary conditions are as close as possible to Dirichlet conditions. In this situation, the currents of matter $I_{2}$ and energy $I_{E}$ may diverge. This situation has been that of our societies since the beginning of the industrial revolution [12], with coal, followed by an acceleration after the Second World War, due to the rise in oil consumption. The divergence of matter and energy currents is directly linked to an increase in the figure of merit, through an increased facilitation of the circulation of matters and energies, which is produced by a minimization of $R_{2}$, as well as an increase of α, i.e., technological progress that allows thermodynamic potentials to be more strongly coupled. A basic illustration of this increase is the performance of steam machines, which have gradually increased the ratio between outlet pressures and inlet temperatures [29]. The second category of system concerns the case where $R_{1}\neq 0$. These systems are particular in that they consume energy, even in a resting situation. We can include living organisms and societies, but also machines, when the latter operate in the idle position, with no other power production than the maintenance of this idle. We have seen that, in this case, there are two categories of systems depending on whether we favor power production or low consumption at rest. These two categories are resolutely distinct and it is illusory to think of a system capable of producing a very high power, while maintaining a very low basic consumption. The choice of $R_{1}$, i.e., the dissipation at rest, is also decisive in the dissipation produced by feedback. The issues of minimization or maximization of efficiency and power are therefore part of a much broader framework than initially thought.

## 7. Discussion

We proposed a generic thermodynamic system model that allows for considering several situations of coupling of the energy and matter currents, as well as their conversions. At the local level, the intrinsic performance of the device that constitutes the core of the system was studied. It appears that the best intrinsic performance in terms of power and efficiency is obtained for the devices with the largest figure of merit, without specifying the respective contributions of the conductivities associated with the transport of energy or matter. However, the sensitivity of these devices to changes in the reduced current *j* shows that the intrinsically most efficient devices are also the most constraining because they require precise control of this reduced current, and therefore of energy and matter currents. At the scale of a complete system, the coupling to the external environment very strongly modifies the conclusions compared to the observations made at the local level. It is observed that behavior is mainly governed by the boundary conditions that connect the local system to the resource and the waste. The presence of boundary conditions such as Dirichlet or Neumann leads to a wide variety of behaviors. The ideal Dirichlet conditions are the only ones that do not lead to any feedback, and consequently conduct in the absence of limitations for the energy and matter currents. When the boundary conditions are between Dirichlet and Neumann, many possibilities then arise. The presence or absence of a resting point for the system strongly influences these possibilities in terms of power, but also in terms of waste production associated with the completion of a task. The concept of coefficient of energy cost, COE, is introduced, generalizing the classical COT already established for biological systems. Finally, it is observed that the internal dissipation produced by the presence of $R_{2}$ is always detrimental for both the efficiency and the power. Its only positive contribution is limited to cases where dissipation and entropy production are explicitly sought, as in the case of homeothermic animals.

## Figures and Tables

**Figure 1 entropy-22-00029-f001:**
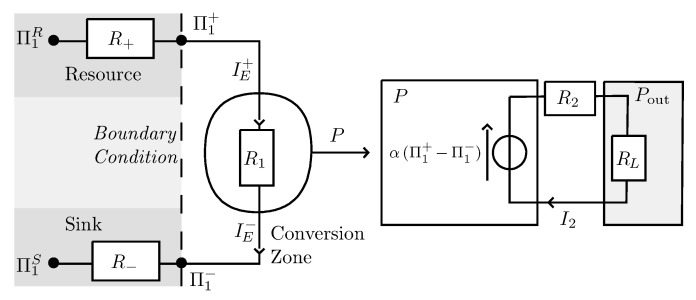
Schematic view of the generic system, with a resource and a sink, whose potential $\Pi_{1}^{R}$ and $\Pi_{1}^{S}$ are constant. The coupling of the conversion zone (circle) with the two reservoirs is ensured by the elements $R_{+}$ and $R_{-}$. As a result, the difference potential $\Pi_{1}^{+}-\Pi_{1}^{-}$ is less than that between reservoir and sink. Power produced in the conversion zone (circle) is $P=-\alpha \Delta \Pi_{1}=\Delta \Pi_{2}$. The internal resistance $R_{2}=\frac{L}{A\sigma_{2}}$ gives rise to a dissipative contribution $R_{2}I_{2}^{2}$. The $R_{L}$ resistance is the output load, and the output power is $P_{\mathrm{out}}=R_{L}I_{2}^{2}$.

**Figure 2 entropy-22-00029-f002:**
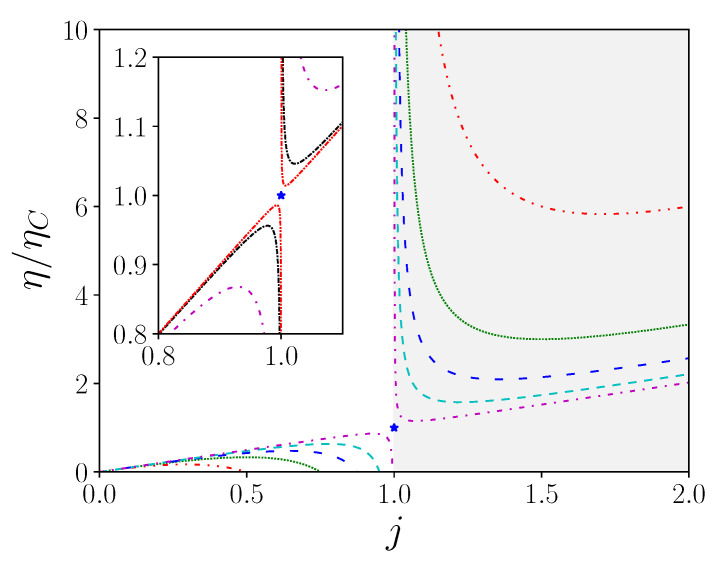
Normalized efficiency $\frac{\eta }{\eta_{C}}$ according to reduced current $j=\alpha J_{2}/J_{1}$ with γ=2 (red, dot dashed), 4 (green, dots), 8 (blue, loosely dashed), 20 (cyan, dashed), $2\times 10^{2}$ (magenta, loosely dot dashed) in main figure, and $\gamma =2\times 10^{2}$ (magenta, loosely dot dashed), $2\times 10^{3}$ (black, dot dashed), $2\times 10^{4}$ (red, dot dot dashed) in inset. The grey area corresponds to the receptor mode (resp. generator mode). Note that the figure is symmetrical with respect to the Carnot point (blue star), which is never reached. This singular point defines the reversible configuration, where causality is broken.

**Figure 3 entropy-22-00029-f003:**
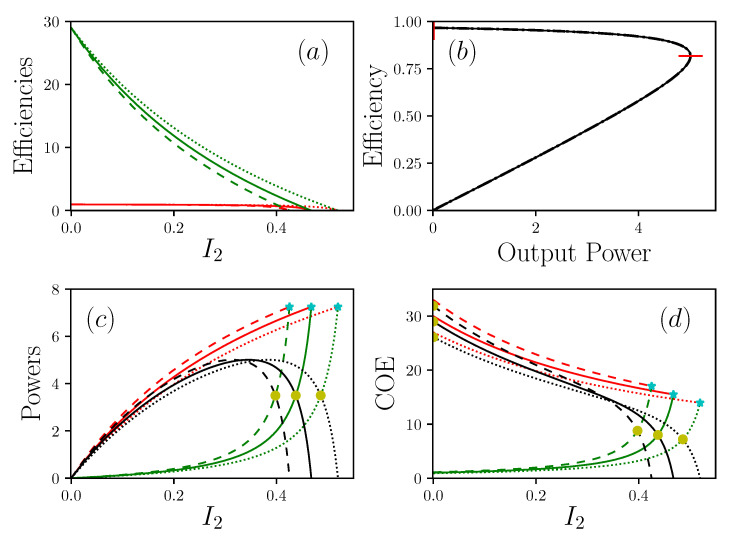
Representations of the powers $I_{E+}$, $I_{E-}$, with $R_{1}^{-1}=0$, $R_{2}=0$ (and $P=P_{\mathrm{out}}$), $R_{+}=R_{-}=2$, $\Pi_{1}^{S}=1$, $\Pi_{1}^{R}=30$. (**a**) shows efficiencies (resp. in red and green) $\eta_{+/-}=P_{\mathrm{out}}/I_{2_{+/-}}$ in function of $I_{2}$, the current of matter. (**b**) is the efficiency in function of the power produced $P_{\mathrm{out}}$. (**c**) show the power (resp. in red, green and black) $I_{E+}$, $I_{E-}$ and $P_{\mathrm{out}}$ in function of $I_{2}$. (**d**) shows (resp. in red, green and black) $COE_{+/-}=I_{E_{+/-}}/I_{2}$ and $COE_{P_{\mathrm{out}}}=P_{\mathrm{out}}/I_{2}$ in function of $I_{2}$. Dotted lines are α=0.9, solid lines are α=1, dashed lines are α=1.1. In (**c**) and (**d**) cyan stars show short circuit situations $I_{2sc}$, yellow circles are critical points $I_{2cp}$. In (**b**) vertical and horizontal red lines are respectively maximal efficiency and maximal power.

**Figure 4 entropy-22-00029-f004:**
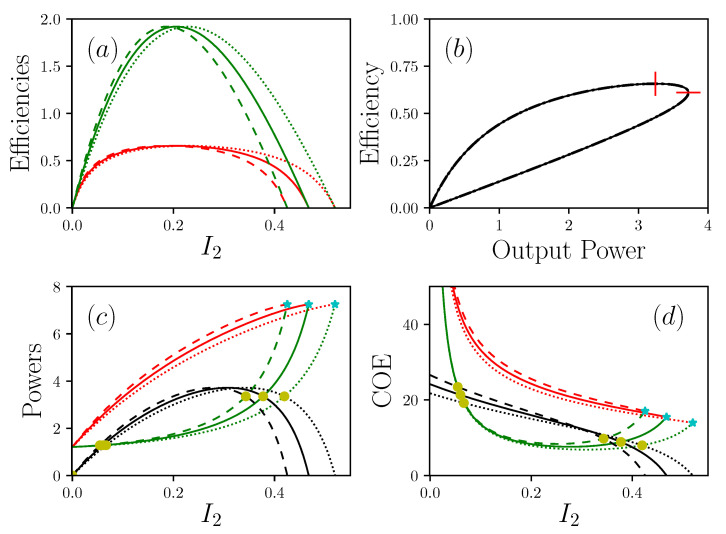
Representations of the powers $I_{E+}$, $I_{E-}$ and *P*, with $R_{1}^{-1}=0.05$, $R_{2}=0$ (and $P=P_{\mathrm{out}}$), $R_{+}=R_{-}=2$, $\Pi_{1}^{S}=1$, $\Pi_{1}^{R}=30$. (**a**) shows efficiencies (resp. in red and green) $\eta_{+/-}=P_{\mathrm{out}}/I_{2_{+/-}}$ in function of $I_{2}$ the current of matter; (**b**) is the efficiency in function of the power produced $P_{\mathrm{out}}$; (**c**) shows the power (resp. in red, green and black) $I_{E+}$, $I_{E-}$ and $P_{\mathrm{out}}$ in function of $I_{2}$; (**d**) shows (resp. in red, green and black) $COE_{+/-}=I_{E_{+/-}}/I_{2}$ and $COE_{P_{\mathrm{out}}}=P/I_{2}$ in a function of $I_{2}$. Dotted lines are α=0.9, solid lines are α=1, dashed lines are α=1.1. In (**c**) and (**d**), cyan stars show short circuit situations $I_{2sc}$, and yellow circles are critical points $I_{2cp}$. In (**b**), vertical and horizontal red lines are respectively maximal efficiency and maximal power.

**Figure 5 entropy-22-00029-f005:**
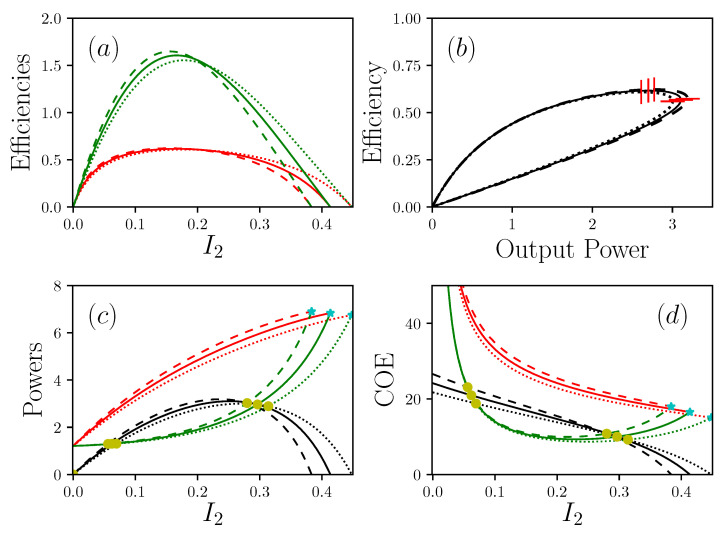
Different representations of the powers $I_{E+}$, $I_{E-}$ and *P*, with $R_{1}^{-1}=0.05$, $R_{2}=4$, $R_{+}=R_{-}=2$, $\Pi_{1}^{S}=1$, $\Pi_{1}^{R}=30$. (**a**) shows efficiencies (resp. in red and green) $\eta_{+/-}=P/I_{2_{+/-}}$ as a function of $I_{2}$ the current of matter. (**b**) is the efficiency in function of the power produced *P*. (**c**) shows the power (resp. in red, green and black) $I_{E+}$, $I_{E-}$ and $P_{\mathrm{out}}$ in function of $I_{2}$. (**d**) shows (resp. in red, green and black) $COE_{+/-}=I_{E_{+/-}}/I_{2}$ and $COE_{P_{\mathrm{out}}}=P_{\mathrm{out}}/I_{2}$ as a function of $I_{2}$. Dotted lines are α=0.9, solid lines are α=1, dashed lines are α=1.1. In (**c**) and (**d**), cyan stars show short circuit situations $I_{2sc}$ and yellow circles are critical points $I_{2cp}$. In (**b**), vertical and horizontal red lines are respectively maximal efficiency and maximal power.

**Figure 6 entropy-22-00029-f006:**
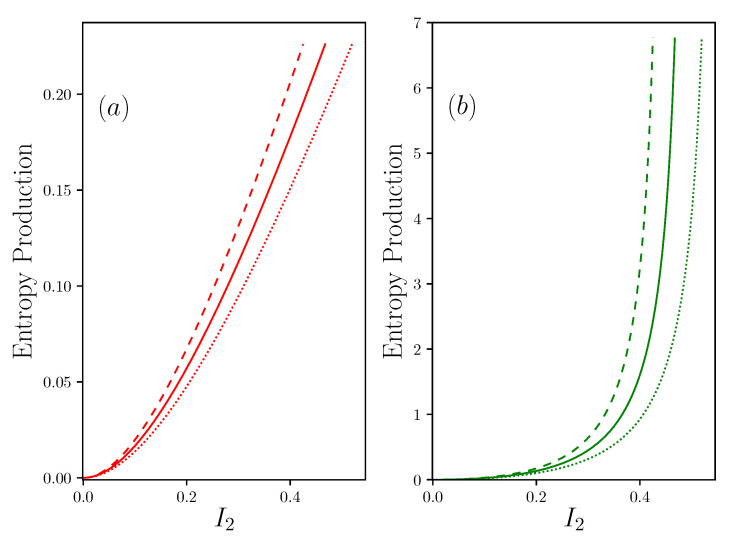
Evaluation of the entropy production with the same configuration as in Figure 3, $R_{1}^{-1}=0$, $R_{2}=0$, $R_{+}=R_{-}=2$, $\Pi_{1}^{S}=1$, $\Pi_{1}^{R}=30$. (**a**) shows ${\dot{S}}_{E+}$ and (**b**) shows ${\dot{S}}_{E-}$, both in function of $I_{2}$ the current of matter—the same color and line-style code as in Figure 3.

**Figure 7 entropy-22-00029-f007:**
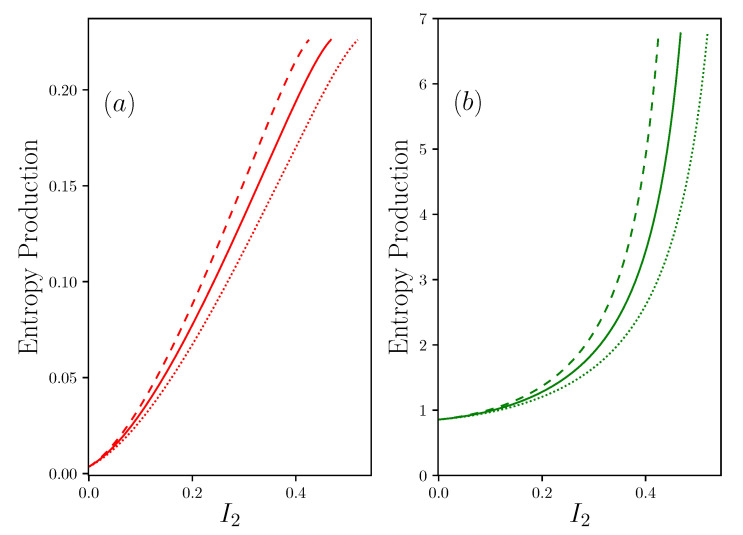
Evaluation of the entropy production with the same configuration as in Figure 4 with $R_{1}^{-1}=0.05$, $R_{2}=0$, $R_{+}=R_{-}=2$, $\Pi_{1}^{S}=1$, $\Pi_{1}^{R}=30$. (**a**) shows ${\dot{S}}_{E+}$ and (**b**) shows ${\dot{S}}_{E-}$, both in function of $I_{2}$ the current of matter—the same color and line-style code as in Figure 3.

**Figure 8 entropy-22-00029-f008:**
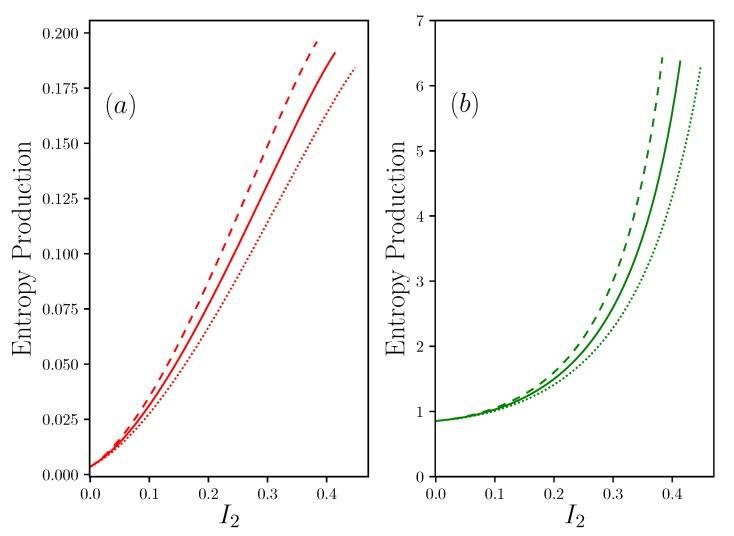
Plot of the entropy production with the same configuration as in Figure 5 with $R_{1}^{-1}=0.05$, $R_{2}=4$, $R_{+}=R_{-}=2$, $\Pi_{1}^{S}=1$, $\Pi_{1}^{R}=30$. (**a**) shows ${\dot{S}}_{E+}$ and (**b**) shows ${\dot{S}}_{E-}$, both in function of $I_{2}$ the current of matter—the same color and linestyle code as in Figure 3.

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
