# Peer review of "Adapted or Adaptable: How to Manage Entropy Production?"

_entropy, 2019, doi:10.3390/e22010029_

Round 1
Reviewer 1 Report
In this work, the authors present an excellent and quality research and therefore deserve to be published in Entropy Journal. In particular, they show two critical facts:
The most efficient system is the most constrained (of course in the context of the present discussion) The best figure of merit necessary leads to a vanishing production of power.
The second point is very critical for the research in quantum transport, where the use of local Osanger matrix formulation is the standard way to calculate, for example, conductance and Seebeck coefficients.
Can this result be applied to that research?
Mean that, therefore, is an erroneous search only to try to increase the figure of merit if the result of the power that is extracted will be zero?
Is this then an analog to the case where the efficiency of an ideal system is calculated obtaining zero power because the idealized processes are of infinite duration?
Author Response
"Please see the attachment."

Reviewer 2 Report
The manuscript is of really outstanding quality, well written and highly informative. I strongly recommend its publication once a few minor points will have been addressed.
Specifically:
1. (p. 4, line 152) it would help comment about the origin of the resistive dipoles and their meaning in the model.
2. (p. 4, line 159) a short comment on what "self-sufficient" means in this frame would be useful
3. (p. 5, line 173) there must be a typo in the phrase "As we will see, ... accurate." - maybe a missing verb
4. (p. 15, line 366) I think that power production, not work production, cancels out. Please either fix the typo or provide a more extensive explanation of the statement.
5. (p. 15, line 376) from -> form
6. (p. 15, line 380) The statement in italics might need rephrasing. Not very clear to me.
7. (p. 16, lines 386-387) please fix the sentence
8. (p. 18, line 428) This -> this
9. all references are missing journal titles, volume and page. Unless this is the format required by "Entropy" (which I don't think) please add them
When these minor points will have been cleared, the paper will be ready for acceptance.
Author Response
"Please see the attachment."
